



# Enhancing the capabilities of a portable FTIR spectrometer for greenhouse gases measurements by addition of a second detector channel for XCO observations

F. Hase(1), M. Frey(1), M. Kiel(1), T. Blumenstock(1), R. Harig(2), A. Keens(2), and J. Orphal(1)

(1) Karlsruhe Institute of Technology (KIT), Institute for Meteorology and Climate Research (IMK-ASF), Karlsruhe, Germany

(2) Bruker Optik GmbH, Ettlingen, Germany

Correspondence to: Frank Hase (frank.hase@kit.edu)

## Abstract

The portable FTIR (Fourier-Transform InfraRed) spectrometer EM27/SUN dedicated for the precise and accurate observation of column-averaged abundances of methane and carbon dioxide has been equipped with a second detector channel which allows the detection of additional species, especially carbon monoxide. This allows an improved characterisation of observed carbon dioxide enhancements and makes the extended spectrometer especially suitable as a validation tool of ESA's Sentinel 5 Precursor mission, as it now covers the same spectral region as used by the infrared channel of the TROPOMI (TROPOspheric Monitoring Instrument) sensor. The extension presented here does not rely on a dichroic, but instead a fraction of the solar beam is decoupled near the aperture stop of the spectrometer using a small plane mirror. This approach allows to maintain the camera-controlled solar tracker setup, which is referenced to the field stop in front of the primary detector. Moreover, the upgrade of existing instruments can be performed without altering the optical setup of the




primary channel and resulting changes of the instrumental characteristics of the original
instrument.

## 1   Introduction

The ground based solar absorption FTIR (Fourier Transform InfraRed) technique is capable
of providing highly reliable measurements of column-averaged $CO_2$ and $CH_4$ abundances.
TCCON (Total Carbon Column Observing Network, Wunch et al., 2011) is one of the
established references for the validation of greenhouse gases measuring space sensors
(Lindqvist et al., 2015; Heymann et al., 2015) and has also been used for e.g. model studies of
greenhouse gases sources (Messerschmidt et al., 2013) and observation of local sources
(Lindenmaier et al., 2014). Recently, several investigators demonstrated that portable low
resolution FTIR spectrometers still allow surprisingly precise and accurate measurements of
column-averaged greenhouse gas abundances (Petri et al., 2012, Gisi et. al. 2012, Frey et. al.,
2015). Such devices are a promising supplement to TCCON, for performing measurements at
remote sites, for mobile applications (Klappenbach et al., 2015), and for observations of
dedicated sources and sinks on regional and smaller scales (Hase et al., 2015).
In this work, we describe an instrumental extension of the system described by Gisi et al. in
2012, comprised of a portable FTIR spectrometer based on a pendulum interferometer design
and a camera-controlled solar tracker for ensuring proper alignment of the line of sight on the
solar disc centre (Gisi et al., 2011). This spectrometer is available today as a standard device
from the Bruker Optics company in Ettlingen under the model name EM27/SUN and is used
by many working groups. The observing strategy follows the same approach demonstrated by
TCCON: the near-infrared spectral bands of the greenhouse gases under study are co-recorded
with the 1.27 um band of molecular oxygen. Thereby, the column-averaged dry air mole
fractions of a target species can be derived from the ratio of the target species column divided
by the molecular oxygen column. This process takes advantage of our accurate knowledge of
the dry air mole fraction of molecular oxygen and helps to reduce various error sources
(Wunch et al., 2011). In addition, TCCON uses DC-coupled interferograms for improving the
precision of the measurements and for quality flagging (Keppel-Aleks et al., 2007), which is
also used for the portable spectrometers. The main difference is that the spectral resolution of
the portable spectrometer is much lower: it provides double-sided interferograms with a



maximum optical path difference (MPD) of 1.8 cm, while TCCON applies 45 cm or higher
MPD. An instructive example of application is provided by Hase et al., 2015, where a set of
five EM27/SUN spectrometers has been used for detecting the carbon dioxide enriched plume
generated by the major city Berlin. The current EM27/SUN spectrometer offers a spectral
coverage of about 5500 to 11000 cm$^{-1}$, slightly broader than the 6000 to 11000 cm$^{-1}$ coverage
used for the prototype described by Gisi et al., 2012. This still conservative choice has been
made to avoid any compromise of measurement accuracy of the primary target variables
$XCO_2$ and $XCH_4$. Nevertheless, it would be highly desirable to add the capability of
observing XCO, which is a highly valuable tool for the characterisation of sources connected
to observed $XCO_2$ enhancements (Wunch et al., 2009). Moreover, an extension of the spectral
coverage of the current EM27/SUN including the 2.3 µm region used by TROPOMI for the
observation of carbon monoxide and methane would qualify the mobile spectrometer as a
validation instrument for the Sentinel 5 Precursor mission.  Here, we introduce an
enhancement of the EM27/SUN by adding the capability of measuring XCO. In section 2 we
provide a summary of the basic design considerations, in section 3 we describe the practical
implementation of the extension, in section 4 we present the characteristics of lamp and
atmospheric solar spectra recorded with the dual-channel prototype, and in section 5 we
demonstrate the performance of the novel setup based on retrievals of atmospheric
observations. Section 6 provides a summary of this study and an outlook towards planned
future activities.

## 2    Design considerations for the XCO extension

The most straightforward approach for an XCO extension is obviously the replacement of the
standard InGaAs diode covering 5500 – 11000 cm$^{-1}$ by a detector element offering extended
spectral coverage. However, wider spectral bandwidth implies increased spectral noise levels
and – more severely – higher susceptibility for a nonlinear detector response. In addition, a
wider spectral response implies that different kinds of out-of-band artefacts, as nonlinearity
and double-passing are superimposing with the wanted spectral signal. The application of an
extended InGaAs diode has been investigated (J. Hedelius et al., 2016), but resulted in a
significant dependence of $XCO_2$ and $XCH_4$ results from the overall signal level. This



characteristic, which damages the reliability of the EM27/SUN primary data products, is
highly undesirable and is not observed with the standard detector element.
An alternative approach is the use of an additional detector element for widening up the
covered spectral region. This avoids the aforementioned problems, but requires either (1) the
use of a sandwich detector element, or (2) alternating observations with different detectors, or
(3) the distribution of the optical beam for feeding two separated detector elements at a time.
A sandwich detector comprised of two sensors with different spectral coverage is a delicate
item of limited availability and considerably higher cost than a pair of separate detectors.
Moreover, the increased number of stacked substrate interfaces promotes the occurrence of
optical resonances, which generate undulations in the spectral domain ("channeling"), a very
nasty feature from the viewpoint of quantitative spectral analysis of crowded spectral scenes
as provided by the atmosphere. The second option of alternating observations is accompanied
by the drawback that species recorded in the two channels are not recorded simultaneously, it
reduces the duty cycle of the measurement, and it requires an additional moving optical
element in the detector branch of the spectrometer, which gives rise to further risks, as
variable misalignment or complete failure of the unit. Therefore, the third approach seems
most promising. This approach has also been realized in the TCCON spectrometer operated
by KIT near Karlsruhe: here, a dichroic allows simultaneous observation of the shortwave
part of the spectral region (covering $O_2$, $CO_2$ and $CH_4$) together with either the longwave part
(covering HF, $N_2O$, and CO) for achieving the same spectral coverage as the standard
extended InGaAs diode used by other TCCON sites, or a spectral section in the mid-infrared
spectral region as observed by spectrometers of the NDACC (Network for the Detection of
Atmospheric Composition Change). The setup uses the same InGaAs detector as the
EM27/SUN for the shortwave and a liquid-nitrogen cooled InSb detector for the longwave
part of the spectrum. Further details and results achieved with this setup are provided by Kiel
et al., 2015a, and Kiel et al. 2015b. The remaining drawback of the approach is that a specific
dichroic is required which complicates the alignment of the detector branch and generates
undulations of the spectral sensitivity. The presence of undulations requires special attention
in the processing of the spectra (Kiel et al., 2015a). In case of the Karlsruhe high-resolution
setup, any loss of the interferometric etendue permitted by the hardware configuration is
undesirable in the InSb branch, because the NDACC-type of measurements are performed
with narrow optical filters at very high spectral resolution. Therefore, the choice of using a
dichroic of good efficiency seems justified. From the viewpoint of the EM27/SUN extension,



the situation differs, as the interferometric etendue is deliberately limited by an adjustable iris
acting as aperture stop. The detector uses only a small fraction of the etendue supported by
the interferometer and solar tracker hardware, and a loss of signal can easily be compensated
by a slight adjustment of the iris aperture. Therefore, in this case a wave front division seems
the most appropriate approach. If finally this wave front division is executed near the aperture
stop of the interferometer (defined by an adjustable iris), the characteristics of the existing
detector branch remain unimpaired and with proper geometrical arrangement of the extension
setup even the functionality of the camera-controlled solar tracker, which references the solar
image to the position of the field stop aperture in front of the existing detector, is maintained.
## 3    Technical setup of the prototype
Figure 1 shows a schematic sketch of the partial beam decoupling and the longwave detector
branch. Behind the adjustable iris aperture which defines the aperture stop of the system, an
off-axis paraboloidal mirror is located which offers 127 mm effective focal length and
generates a solar image on the field stop in front of the existing detector element. Physically,
the field stop is realized by a circular hole of 0.60 mm diameter in a thin disc of stainless
steel. For partial decoupling of the beam, a plane mirror of 10 x 20 mm$^2$ size has been added.
This mirror is located directly behind the off-axis paraboloid and accepts about 40% of the
incoming converging beam. The deflection angle between the residual parent and reflected
partial beam amounts about 25 degree and the deviation is chosen along the horizontal, so that
the second detector element can be mounted next to the original detector. The solar image is
formed on a secondary aperture stop (0.80 mm diameter) in front of the additional detector
element. The detector element used is a windowless extended InGaAs diode (cut off 4000 cm$^-$
$^1$) offering 1 mm$^2$ sensitive area. In a gap between the secondary field stop and the diode a
wedged Ge long pass filter is mounted which shields the extended InGaAs diode from the
spectral section already covered by the primary detector. Figure 2 - 4 show close-up
photographs of the dual-detector prototype. Figure 2 shows the small plane mirror just in front
of the off-axis paraboloid. The mirror is glued with a two-component epoxy resin adhesive to
its aluminium support. The bonding layer has been chosen thick and flexible enough to avoid
deformation of the glass mirror due to temperature changes. The support structure allows fine
adjustment of the direction of the beam reflected towards the secondary field stop. Figure 3



shows the image of an artificial source on the primary and secondary field stop. Figure 4
shows the detectors from the top. While the primary detector and preamplifier unit remain in
the standard detector box, a short cable is used for operating the secondary detector outside of
its box, which houses the preamplifier unit only. The secondary detector is mounted on a
support structure which in turn is solidly fixed to the primary detector box. The alignment of
the secondary fieldstop is performed by using an artificial external light source. The solar
tracker is used for conveniently centring the image of the source on the primary fieldstop, the
evaluation is performed with the camera of the solar tracker. The fine adjustment of the
source image on the secondary field stop is performed by aligning the plane mirror, the
evaluation is performed by eye with a magnifier. Finally, the position of the unit containing
the wedged Ge filter and the secondary detector element is adjusted with respect to the
secondary field stop by searching for maximum signal level.
**4   Characteristics of spectra recorded with the dual-channel prototype**
Figure 5 shows a lamp spectrum recorded with the prototype. Both the primary and secondary
channels are essentially free from channeling (we estimate the upper limit for the peak-to-
peak amplitude in the primary channel to 0.0005, in the secondary channel to 0.0002) and a
proper definition of the optical bandpasses is achieved: The low wavenumber limit of the
secondary detector results from the cut-off of the diode, its high wavenumber limit is defined
by the Ge filter. Similarly, the low wavenumber limit of the primary detector is shaped by the
diode cut-off, the high wavenumber limit is due to a Schott RG 830 longpass filter glass
which acts as entrance window of the spectrometer. Therefore, both detectors observe through
this optical element and a further extension of the concept presented here towards lower
wavenumbers would in addition to a suited detector element require a replacement of entrance
window. Finally, also a beamsplitter exchange would be needed, because the standard
EM27/SUN is equipped with a Quartz substrate beamsplitter.
Figure 6 shows a raw solar spectrum recorded with the prototype. This spectrum reveals that
the transition region between both channels coincides nicely with the opaque region created
by the strong water vapour absorption between the atmospheric window regions H and K.
Close inspection of the zero baselines reveals a minor out-of band offset of the order of
0.015% for the primary detector branch and no indication of out-of band signal for the



secondary detector branch (below 0.0025%). The 1-sigma signal-to-noise ratio in the peak of
the primary channel is 13 000, the signal-to-noise ratio of the secondary channel reaches 20

3    000.

**5    Results of atmospheric retrievals**
For an evaluation of the instrument performance the modified spectrometer has been operated
in parallel to a standard EM27/SUN for 6 days of measurements from mid of October to mid
of November 2015. Before the implementation of the extension, the selected spectrometer has
been operated side-by-side to the same standard EM27/SUN on various occasions. This
allows us to check whether the modification of the spectrometer changed the oxygen column
derived from the primary channel. We assume that the oxygen column is the most sensitive
indicator for changes of the instrumental characteristics (especially instrumental line shape
(ILS)), as instrumental error sources tend to cancel out in the final column-averaged
abundances of the target gases. For this purpose, we selected seven days of measurements
before the intervention was performed on the spectrometer and six days after the intervention.
Table 1 and table 2 list the measurements used in this study. Table 1 collects the
measurements taken between middle of May and end of August side-by-side with the
spectrometer foreseen for the extension and a standard EM27/SUN used as reference. Table 2
collects the measurements taken after implementation of the dual-channel extension, the
standard EM27/SUN reference (same spectrometer used as before), and the TCCON
spectrometer Karlsruhe. The low-resolution interferograms used in this study passed the
quality check implemented in our calibration routines (based on average DC level and DC
variations during recording), the TCCON data used passed the quality flagging of the GGG
processor. For limiting airmass-dependent effects, data recorded at solar elevations below 15
degrees have been discarded. Note that the TCCON time series is somewhat sparser than for a
typical TCCON site, due to alternating recording of high-resolution mid-infrared spectra.
The EM27/SUN side-by-side observations were performed on the roof terrace of our institute,
in a distance of less than 1 km from the TCCON spectrometer site Karlsruhe. In addition, the
TCCON spectrometer was operational during all days of observations with the dual-channel



EM27/SUN. Therefore, the official TCCON XCO product derived from these observations
can serve as the best available reference for the true column-averaged CO abundances. In
addition to the standard operation mode, the high-resolution spectrometer records
intermittently double-sided interferograms with the same resolution as applied by the
EM27/SUN, which are very useful to evaluate systematic retrieval biases introduced by the
significantly different resolution of the TCCON spectrometer and the EM27/SUN.
For the pre-processing of the EM27/SUN raw data, our suite of CALPY routines was used
(Frey et al., 2015). For the analysis of the EM27/SUN we applied the retrieval code PROFFIT
(Hase et al., 2004). PROFFIT is in wide use in the NDACC (e.g. Sepúlveda et al., 2014;
Virolainen et al., 2014; Mengistu Tsidu et al.; 2014), is in very good agreement with the
official TCCON analysis (Dohe, 2013) and is also successfully applied for the analysis of
spectra recorded with the EM27/SUN (Gisi et al., 2012; Frey et al., 2015). For the CO
analysis of the EM27/SUN spectra, the pressure-temperature, methane, water vapour, HDO,
and CO profiles have been adopted from the TCCON processor. Figure 7 shows a typical
spectral fit for the spectral window 4210 to 4320 $cm^{-1}$ which is used for the CO analysis.
Systematic fit residuals significantly larger than the noise level of the spectrometer are
evident, indicating that the simulation of this crowded spectral scene shaped by numerous
strong absorption lines could be improved by further progress on spectroscopic data (our fits
are essentially based on HITRAN 2008, see Rothman et al., 2008). The provision of an
improved and consolidated set of spectroscopic data for the spectral region under
consideration here is an ongoing effort (e.g. Galli et al., 2012; Scheepmaker et al., 2013;
http://seom.esa.int/page_project003.php).
Before we discuss results derived from the new spectral channel, we investigate whether the
modifications performed on the prototype affected the results for the oxygen column, which is
derived from the existing spectral channel. Figure 8 compares the oxygen column from the
dual-channel prototype before and after the intervention and the standard EM27/SUN used as
a reference. The agreement is excellent, no detectable offset as result of the dual channel
implementation is found. We therefore assume that the implementation of the extension did
not affect the behaviour of spectrometer's primary channel. Note that a comparison of oxygen
columns is a very sensitive test, as many instrumental errors tend to cancel out in the final
column averaged abundances of the target species.





Next, we investigate the compatibility of $XCH_4$ derived from the primary detector with the
$XCH_4$ from the secondary detector. $CH_4$ is the main absorber in the $4200 - 4320$ cm$^{-1}$ region,
followed by $H_2O$ and HDO. The CO overtone band is weak in comparison to the signatures of
the other species, so $CH_4$ provides a more sensitive handle for revealing any instrumental
issues. Figure 9 shows the $XCH_4$ from both channels of the prototype. The two datasets are in
excellent agreement. There is a systematic offset between the two time series (scaling factor
1.01103), very likely due to residual inconsistencies of the spectroscopic line intensities. The
standard deviation of the ratio of both time series is 0.092%. It should be noted that a
misalignment between the primary and secondary fieldstop would induce systematic
differences as function of azimuthal viewing angle. Indeed, the comparison of the two $XCH_4$
products provides a tool for detecting a misalignment of the secondary fieldstop, because a
variation induced by our limited capability to simulate the spectral scene will be symmetric
around local noon, whereas the fieldstop misalignment will differ between morning and
afternoon as a consequence of the rotation of the image on the fieldstop as function of
azimuthal viewing angle (Reichert et al., 2015). At a solar elevation angle of 20° a 0.1°
displacement of the solar disc would create an error of up to 0.5% in dry air mole fractions
derived from the secondary channel (solar elevation 10°: up to 1%), so this effect can easily
be detected using the two methane data products and subsequently be applied for establishing
a correction for XCO, if required. However, we do not observe such kind of a suspicious
discrepancy between the two time series.
Finally, we investigate the XCO timeseries derived from the prototype. Figure 10 displays the
whole dataset as function of airmass, the daily mean values have been scaled in order to
remove the day-to-day variability. Obviously, the results suffer from an airmass dependency.
The presence of such kind of artificial airmass dependency is a frequent problem created by
our inability of simulating the observed spectral scene perfectly well. We therefore apply a
second-order polynomial (shown in figure 10) for removing this artefact. We have located the
neutral point of this correction at 25 degree solar elevation angle, which is near the average
solar elevation angle of the complete dataset. Note that TCCON also requires the aid of
empirical airmass corrections for carbon monoxide and other target species. Figure 11
presents the XCO as derived from the dual-channel EM27/SUN prototype in comparison to
the official TCCON data product. The airmass correction clearly improves the agreement with
the TCCON reference. The airmass-corrected XCO data deduced from the dual-channel
prototype are slightly smaller (scaling factor 0.978) than the TCCON reference and the





standard deviation of the difference of the daily means is 1.1 %. Note that apart from the
empirical airmass correction and the TCCON calibration factor derived from aircraft
measurements (Kiel et al., 2015b) no adjustments on the XCO derived from the prototype
have been applied. In our feeling, this result is quite satisfying, and we expect that the
upcoming set of improved spectroscopic data will further improve the consistency of XCO
results between TCCON and the low-resolution measurements. In October, 12, 2015, a larger
intraday variability of XCO occurred, which has been nicely sampled by both the TCCON
spectrometer and the prototype. The results are shown in figure 12, indicating that the
prototype can detect XCO enhancements on the 0.5 % level.
## 6   Summary and Outlook
The portable EM27/SUN spectrometer is dedicated to measurements of column-averaged
abundances of carbon dioxide and methane with sufficient quality for climate research. In this
work, we have described a dual-channel extension of this device which can be added in a
straightforward manner without interfering with the standard setup of the spectrometer. The
second channel uses an extended InGaAs detector element and a wedged Ge filter to define a
spectral bandpass beyond the spectral coverage of the standard spectrometer. It is fed via a
small plane mirror which decouples parts of the beam towards a secondary detector element.
This approach avoids interference with the concept of a camera-controlled tracker referenced
to the fieldstop and conserves the spectral coverage and characteristics of the primary
detector. The secondary detector allows the simultaneous measurement of XCO, which is a
very useful tool for source characterisation, while the use of a cooled detector element can be
avoided. In the second part of this work, we performed a preliminary validation of the setup
by verifying that the dual-channel prototype maintains the characteristics of the standard
spectrometer and by investigating the XCO data product. We pointed out that due to the fact
that methane is the primary absorber in the spectral window used for the carbon monoxide
analysis, the comparison of $XCH_4$ derived from either the primary or the secondary channel
can be used as a diagnostic tool for detecting a residual misalignment of the secondary
fieldstop or other instrumental issues.
We plan a further in-depth characterisation of the dual-channel EM27/SUN prototype
introduced here. The spectrometer is foreseen to participate in the TCCONcomp campaign



funded by ESA. This campaign is scheduled for 2016, is led by University of Bremen,
Germany, and BIRA (Royal Belgian Institute for Space Aeronomy), Belgium, and involves
several partners contributing various promising approaches for the Sentinel 5 Precursor
validation. Aim of TCCONcomp is to validate these techniques with respect to the TCCON
reference instrument operated by FMI (Finnish Meteorological Institute) at the Sodankyla
station, Finland.
We would like to mention that the secondary channel of the extended EM27/SUN also allows
observation of HDO, opening up the possibility of using the device also for observations of
water vapour isotopic variability, although the verification of the quality and information
content of such data from low-resolution spectra will require careful validation efforts (for
prior work based on high-resolution near-infrared spectra see Rokotyan et al., 2014). The new
spectroscopic datasets under preparation in support of the Sentinel 5 Precursor mission will be
a highly valuable ingredient for TCCON, for the spectrometer presented here and for any
other remote sensing devices working in the 2.35 μm spectral region targeting at methane,
carbon monoxide or water vapour isotopic composition.
**Acknowledgements**
We acknowledge support by the ACROSS and MOSES research infrastructures of the
Helmholtz Association.
We acknowledge support by Deutsche Forschungsgemeinschaft and Open Access Publishing
Fund of the Karlsruhe Institute of Technology.





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



1 Table 1. Overview of measurement days after before modification of the spectrometer. The

2 number of spectra used in the comparison is indicated.

| Date (JJMMDD) | Number of spectra recorded with the EM27/SUN selected for modification (S/N0039) | Number of spectra recorded with the EM27/SUN used as a reference (S/N0037) |
|---|---|---|
| 150518 | 368 | 369 |
| 150521 | 282 | 276 |
| 150702 | 465 | 477 |
| 150703 | 472 | 473 |
| 150706 | 342 | 344 |
| 150710 | 338 | 330 |
| 150831 | 387 | 400 |



1    Table 2. Overview of measurement days after modification of the spectrometer. The number

2    of spectra used in the comparison is indicated.

| Date (JJMMDD) | Number of spectra recorded with the modified EM27/SUN (S/N0039) | Number of spectra recorded with the EM27/SUN used as a reference (S/N0037) | Number of spectra recorded with the 125HR spectrometer (high / low resolution spectra) |
|---|---|---|---|
| 151012 | 206 | not operated | 86 |
| 151026 | 186 | 201 | 40 |
| 151105 | 136 | 173 | 72 |
| 151110 | 83 | 114 | 33 |
| 151111 | 103 | 107 | 20 |
| 151116 | 79 | 63 | 71 |





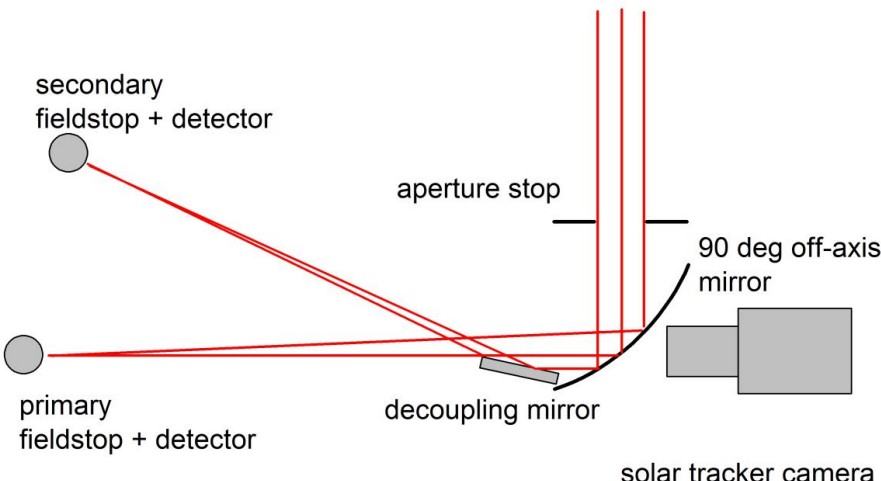

Figure 1. : Schematic drawing of the detector branch of the extended EM27/SUN. The camera
is located above the drawing plane and images the primary fieldstop. For clarity, the camera
position has been shifted to the right in the drawing, it actually is located above the off-axis
mirror.

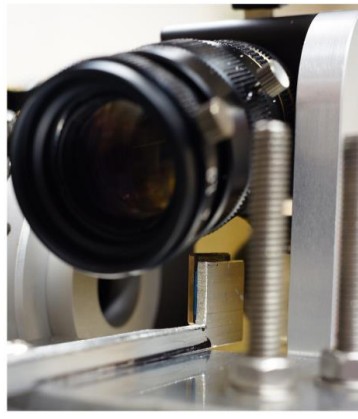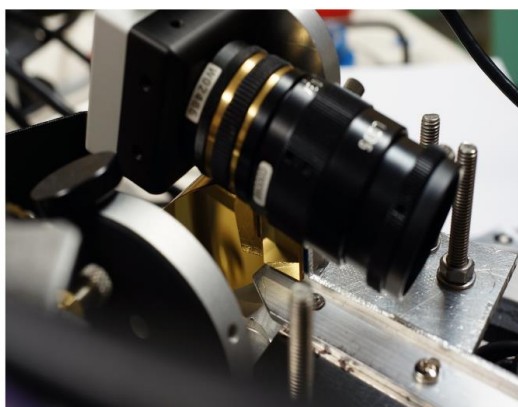

Figure 2. Close-up of the plane mirror used for decoupling of the secondary beam. Figure 2a
(left side) shows the support arm which carries the mirror. The mirror itself is seen from the
back. On the left, parts of the circular mounting of the adjustable iris are seen. Behind the
contours of the plane mirror, the larger off axis paraboloid is located. Above the mirror, the
camera used for the camera-controlled solar tracker is located. Figure 2b (right side) offers a
different perspective: below the camera, the front side of the plane mirror is seen. The frame
of the iris is seen from the back.



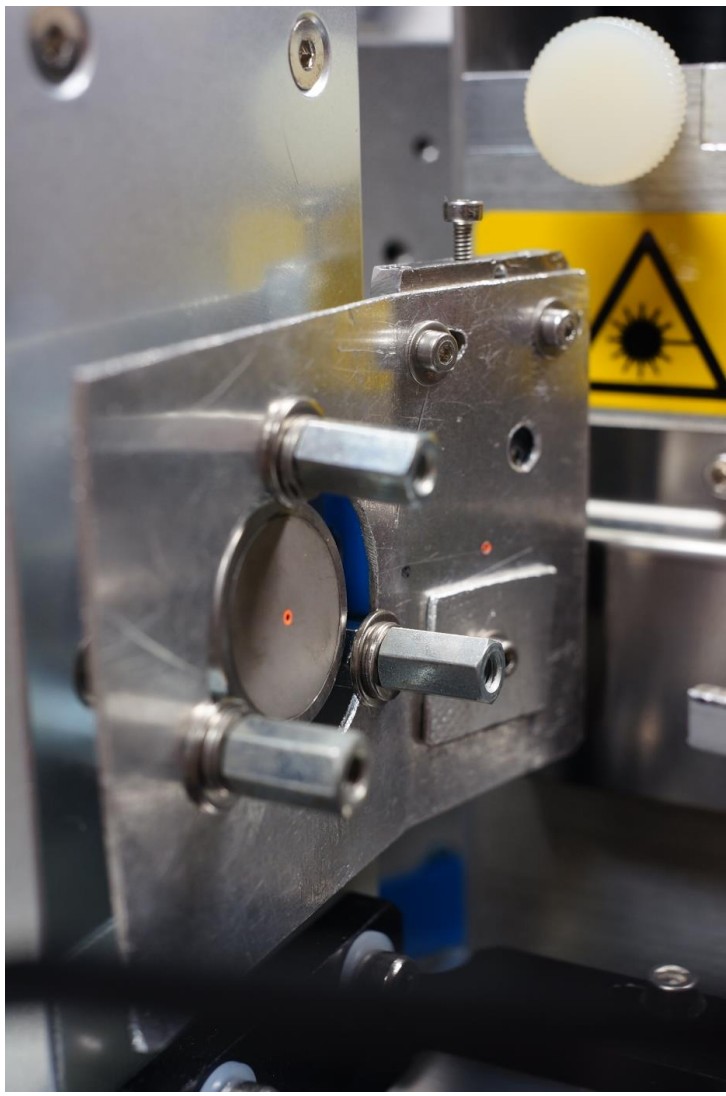

Figure 3. In the front, the image of an artificial source covering about the same angle as the
solar disc is seen on the primary field stop. The support for the secondary field stop is
mounted on the primary detector. The secondary field stop itself is realized as a 0.8 mm hole
in the aluminium sheet. The image of the source on the secondary field stop is seen to the
upper right from the centre of the image. The final alignment of the image position with



1    respect to the position of the secondary field stop is performed by fine adjustment of the plane

2    mirror.




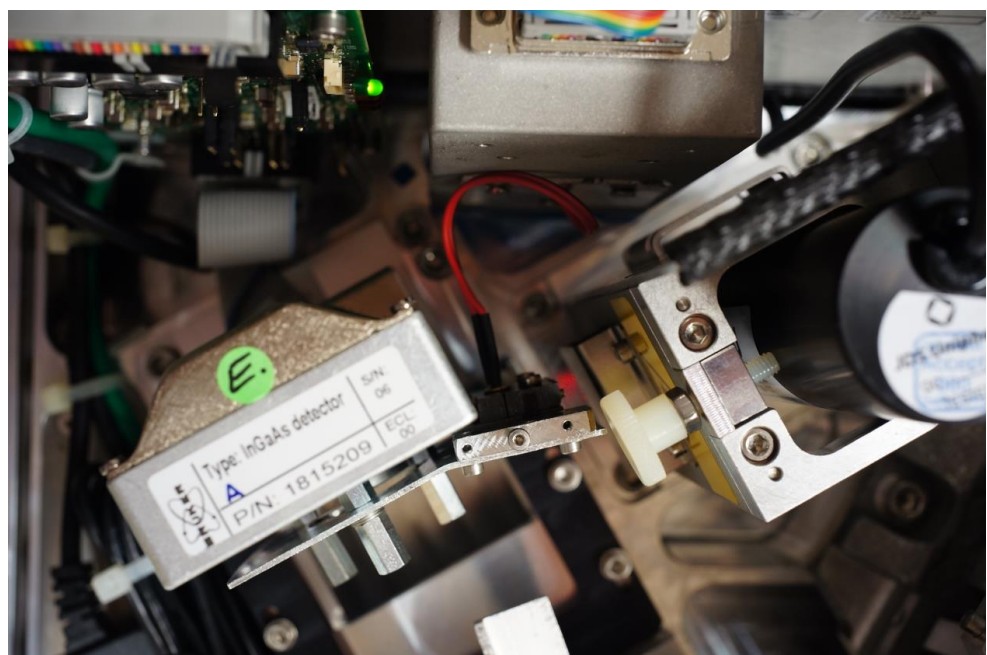

Figure 4. This image provides a top view of the detector units. The primary detector remains in its standard detector box, the supporting structure for the secondary field stop is mounted on the box of the primary detector. The secondary detector and the Ge filter are accommodated in a separate holder, which can be adjusted laterally with respect to the field stop.



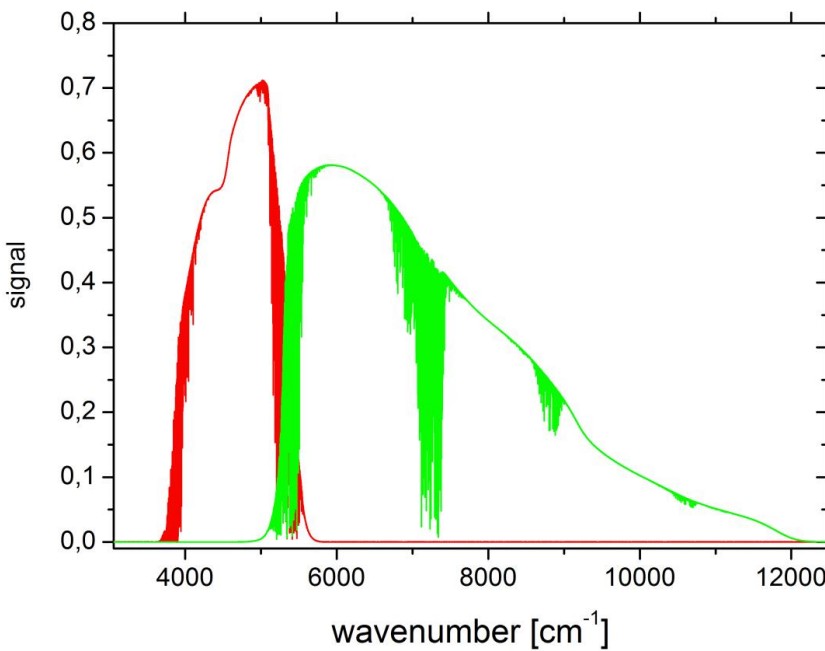

2    Figure 5. Lamp spectrum recorded with the dual channel prototype. The primary detector

3    covers the spectral section observed with the standard EM27/SUN, the secondary detector

4    covers the 4000 – 5500 cm$^{-1}$ region.



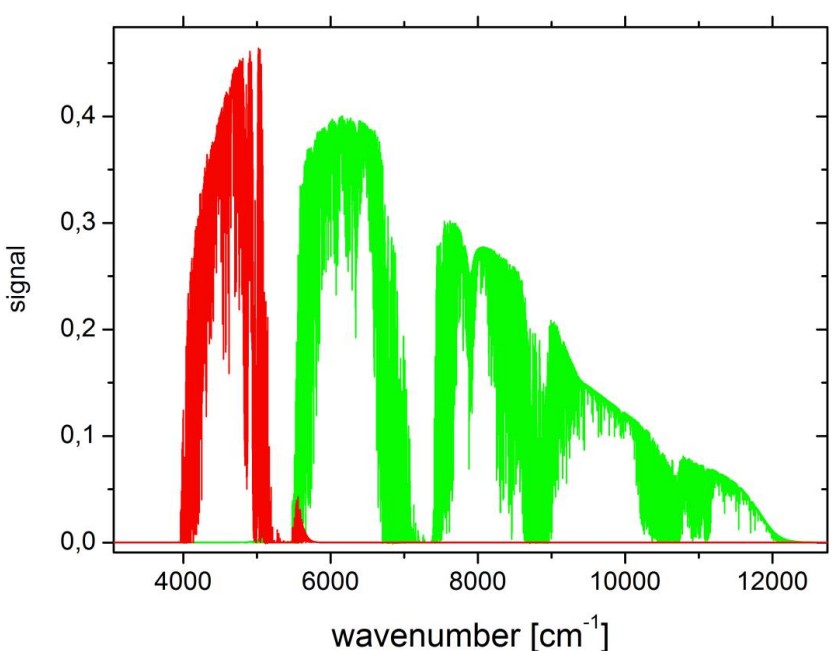

2      Figure 6. Solar spectrum recorded with the dual channel prototype.





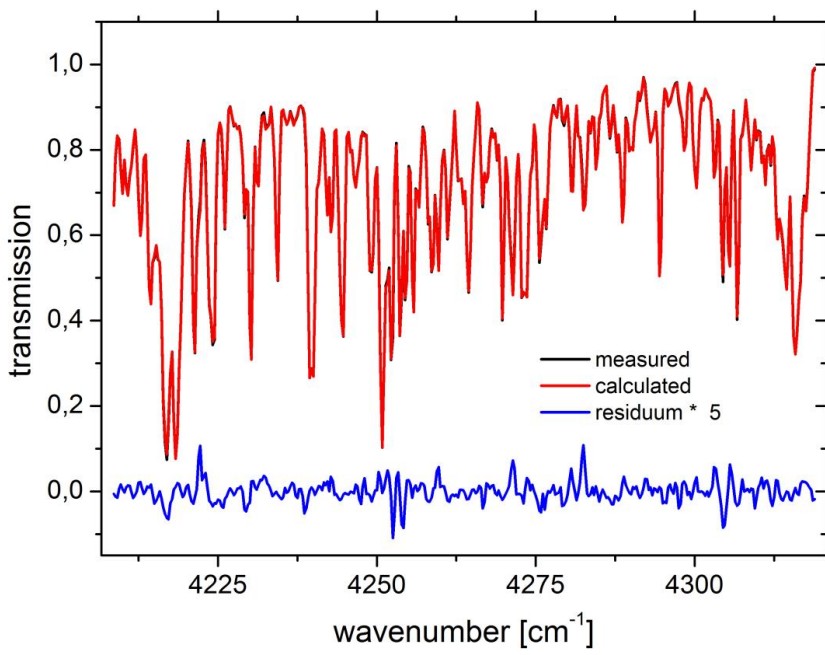

2    Figure 7. Typical spectral fit for the selected CO fitting region (4210 – 4320 cm$^{-1}$). Interfering

3    species are $CH_4$, $H_2O$ and HDO.



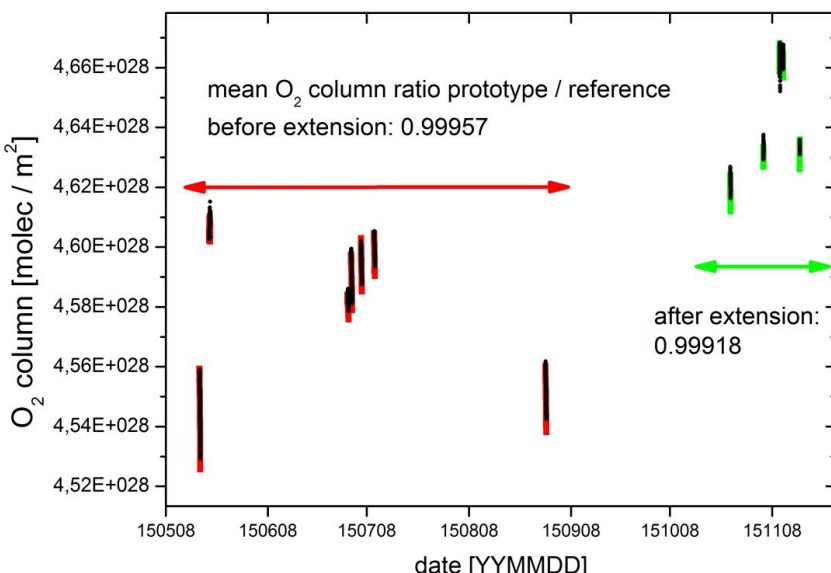

Figure 8. Total column amounts of molecular oxygen measured with the EM27/SUN used as
a reference (data in black) and the prototype before and after the implementation (before: red,
after: green) of the secondary channel. The implemented extension did not affect the
characteristics of the prototype.





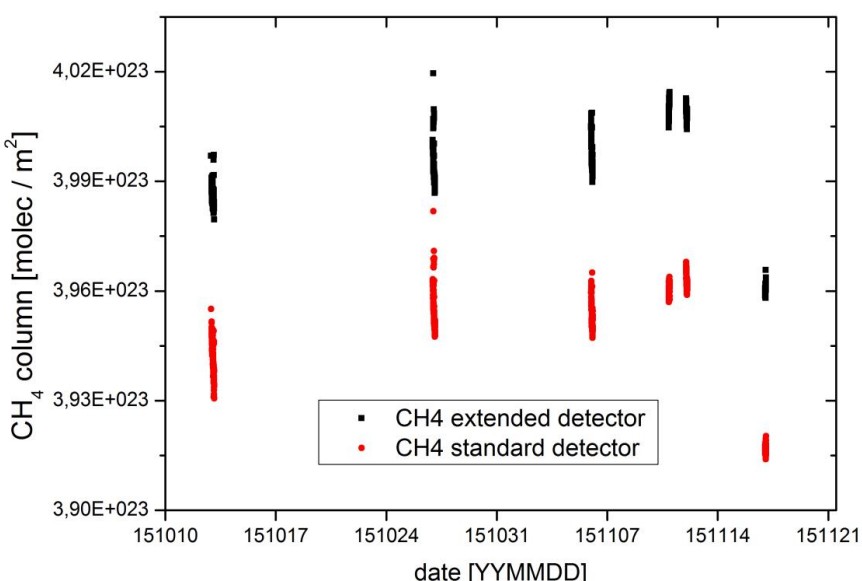

2      Figure 9. Time series of methane as derived from the primary and secondary spectral channel.





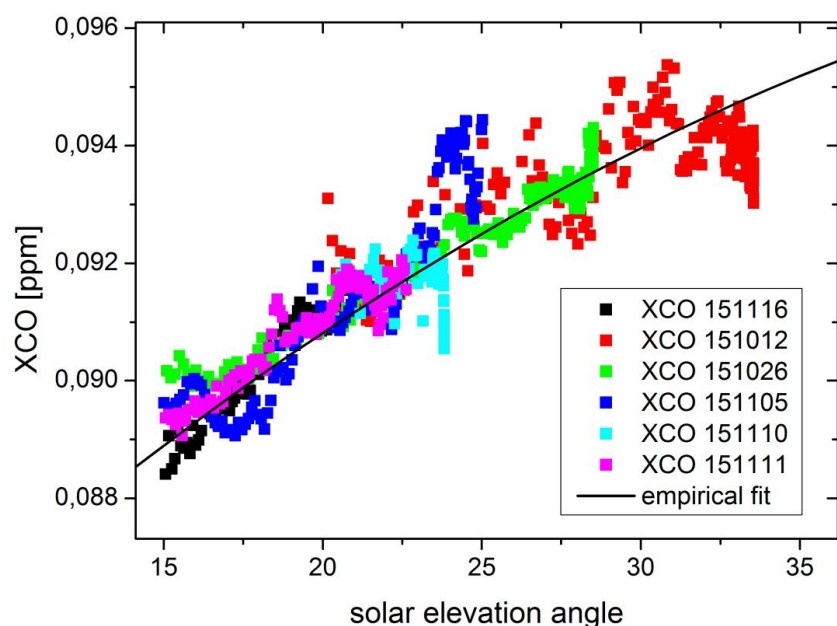

2    Figure 10. Apparent airmass dependency of the XCO data observed with the dual-channel

3    prototype and empirical fit.





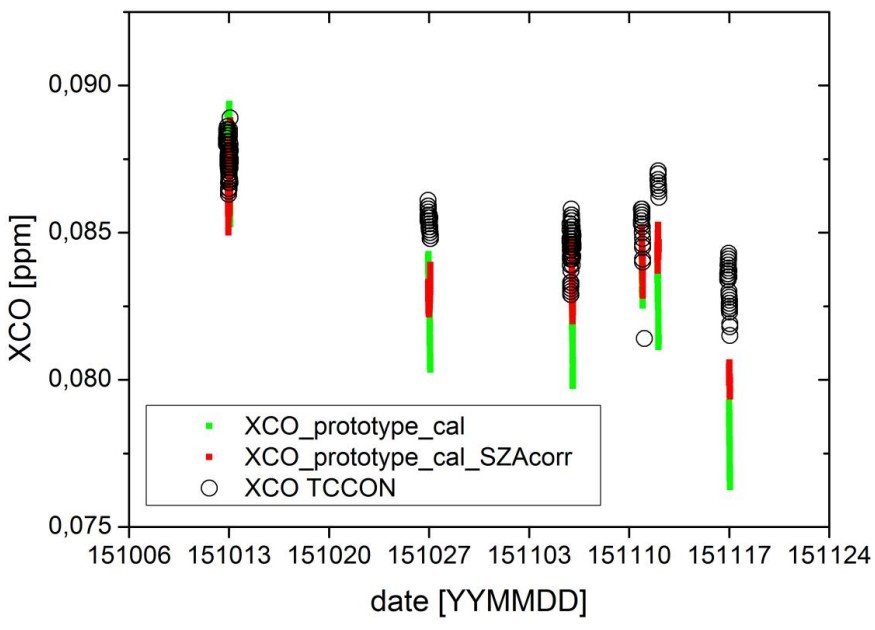

3     Figure 11. XCO deduced from the dual-channel prototype and TCCON site Karlsruhe.


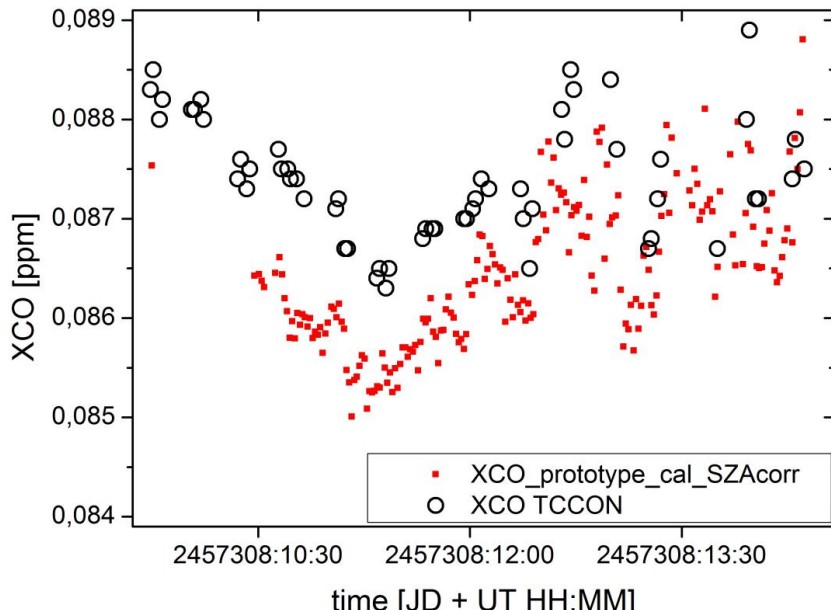

2    Figure 12. XCO intraday variability as observed by TCCON and the dual-channel prototype.

