# Peer review of "Manuscript under review for journal Atmos. Meas. Tech."

_Atmospheric Measurement Techniques, 2015_

## Referee Comment (RC1) · Anonymous Referee #1 · 23 Feb 2016

**1   General Comments**

Recently there has been a push to develop portable instruments capable of measuring total column abundances of greenhouse gases. Perhaps the most well developed (to date) of these is the EM27/SUN. In order to achieve the performance necessary from the instrument, the spectral range of this portable FTIR is limited to a smaller range than the "gold standard" for column greenhouse gases measurements, TCCON. This means that $XCO_2$ and $XCH_4$ measurements are possible, but XCO and other gases that are measured by TCCON are not. The paper by Hase et al describes a modification to the

[Figure]

EM27/SUN instrument that allows simultaneous measurements of CO. They achieve this via addition of a second channel.

Extending the measurement capabilities of the portable instrument is certainly a worthwhile, even desirable, goal. Measuring CO along with $CO_2$ and $CH_4$ would enhance the instrument's capabilities for one of its applications, namely measurements around cities. The additional CO constraint would be particularly valuable for tracing anthropogenic plumes. In addition, it would extend the validation capabilities for satellite missions such as the Sentinel 5-Precursor mission, and GOSAT-2, which plans to also measure CO. An additional advantage of the modifications made by Hase et al is that existing instruments can be updated to measure the extra channel with minimal instrument intervention.

Overall, the paper is reasonably well written, and the description of the instrument, its modifications and the reasoning behind them is certainly sufficient. In some cases, these are perhaps too detailed and could benefit from some simplification. The nature of the article makes me wonder if it would be better suited to Geoscientific Instrumentation, Methods and Data Systems than AMT, but it is certainly also suitable for AMT. I do, however, have some concerns that I would like to see addressed before its publication. These are largely focussed on the comparison to existing techniques, both TCCON and the standard EM27/SUN.

- Throughout there are many subjective and "colourful" terms used that are not strictly scientific. I have attempted to note these in the technical comments, but these should be tidied up.

- There are likewise still a few language issues throughout that would make the article easier to read. In addition, the article features many long paragraphs, and would benefit with respect to clarity if these were broken up.

- The comparisons between the standard measurements and those made using

the prototype are often fairly rudimentary. In my opinion, the plots should also show the differences or ratios, and some more rigorous reporting of comparison statistics is necessary to understand the performance and limitations of the modified instrument. This should be done for all the comparisons reported.

- Similarly, the description of the airmass correction is poorly described. The relative corrected and uncorrected values should be plotted against solar zenith angle, and the residuals with respect to the fitted function shown. The function itself (co-efficients) should be described, and if it fails at zenith angles greater than $75°$ this should also be shown (maybe greyed out). From Figure 10, there also appears to be evidence that the correction might fail at higher elevation angles, though presumably this is due to the variation on that particular day. Further work is obviously necessary to quantify this, particularly if the instrument is to be deployed to instruments in different latitude bands.

- From Figure 11, it seems like there is some serious day-to-day variability in the agreement with TCCON. Please quantify and discuss this further than what is already mentioned.

**2  Technical Comments**

There are numerous places where the article requires copy editing. I have also listed as many technical corrections as possible below:

- title: this should read "greenhouse gas measurements" instead of "greenhouse gases measurements". I would also suggest trying to reduce the number of words in the title, at presents it is quite cumbersome.

- p1, l18: add a comma after 'channel'

- p1, l18-19: you mention additional species here, but the article focussed on CO. I suggest removing "additional species, especially"

- p2, l8: "The TCCON"

- p2, l9: greenhouse gases → greenhouse gas. Also some better references for satellite validation would be good here. The Lindqvist reference in particular doesn't really fit. Seminal references for GOSAT, OCO-2 and/or SCIAMACHY validation using TCCON would be better, such as Reuter et al, 2011, Wunch et al, 2011, Butz et al, 2011, or Morino et al, 2011.

- p3, l1: MPD might be better replaced with OPDmax or MaxOPD, but given that this only seems to be used in this line and the next then there is perhaps no reason to abbreviate it.

- p3, l7: variables → species (or quantities)

- p3, l27: severely → seriously

- p3, l27-29: the sentence spanning these lines need rephrasing for clarity.

- p3, l29-31: I'm not 100% clear what you are trying to say here. Is there a dependence of $XCO_2$ and $XCH_4$ *on* the signal level?

- p4, l3: delete 'up'

- p4, l5: delete the first instance of 'or'

- p4, l6: perhaps replace 'feeding' with 'illuminating'

- p4, l11: replace 'nasty' with something more scientific

- p4, l15: as → such as

- p4, l17-23: this seems like an unnecessarily nepotistic example. Most TCCON sites use a dichroic to measure simultaneously on InGaAs and Si detectors.

- p4, l3- p5, l9: this is a massive paragraph. Please break up into smaller paragraphs.

- p4, l27: add a comma after 'required'

- p4, l29: In case → In the case

- p4, l30: "interferometic etendue" - this might need to be explained for non-optics experts, or could be replaced by a term more suited for laymen.

- p4, l30 - p5, l9: This section uses a lot of words to not say a lot. I would suggest shortening it.

- p5, l14: which → that

- p5, l15: add a comma after located

- p5, l21: amounts → amounts to

- p5, l21: degree → degrees

- p5, l26: define Ge

- p5, l26: add a comma after mounted

- p5, l30: add 'to be' after chosen

- p6, l5: add a comma after structure

- p5, 13 - p6, l12: another paragraph that needs to be broken up into smaller ones.

- p6, l18: (2 instances) add 'be' after 'to'

- p6, l19: what do you mean by 'definition' in this context? I assume you are refer-
ring to the two spectral bandpasses as being well separated and independent.

- p6, l23: which → that

- p6, l23-26: given that you have just said that the entrance window limits at high
wavenumbers, why would this need to be replaced to extend further to lower
wavenumbers?

- p6, l30: for this audience either define regions H and K (e.g. by labelling on
Figure 6), or replace with wavenumber ranges.

- p6, l31-32: how does this figure of 0.015% compare to the offset on the standard
instrument?

- p7, l7: For an evaluation of → To evaluate

- p7, l9-10: has been → was

- p7, l10: to → with

- p7, l17: capitalize the second instance of 'table'

- p7, l17: collects → lists

- p7, l18: middle of May and end of August → mid May and the end of August

- p7, l19: I suggest replace "foreseen for the extension" to "used for the prototype"

- p7, l20: collects → lists (or summarizes')

- p7, l21: suggest changing the words in the parentheses to "the same spectrom-
eter as used previously"

- p7, l22: add 'at' after spectrometer. Also, technically it is not a "TCCON spectrometer", but a FTIR spectrometer, used as part of the TCCON.

- p7, l24: maybe add 'standard' before 'quality flagging'

- p7, l25: maybe replace 'processor' with 'software suite'.

- p7, l25-27: as mentioned in the general comments, it would be good to see evidence for discarding the higher zenith angle spectra. E.g. TCCON includes up to 82 degrees. Limiting the EM27/SUN to 75 degrees would limit the application at higher latitudes.

- p7, l30: 'in a distance' → 'at a distance'

- p8, l3-4: move 'intermittently' to before 'records'

- p8, l5: 'are very useful' → 'can be used'

- p8, l13: of → from

- p8, l15: insert a comma before which

- p8, l27: the comma after excellent should be replaced by a semicolon

- p9, l19: 'such kind of a suspicious' → 'such a'

- p9, l23: 'suffer from' → 'are affected by'

- p9, l24: 'such kind of' → 'such an'

- p9, l25: 'of simulating' → 'to simulate'

- p9, l25: delete 'well'

- p9, l26: 'for removing' → 'to remove'
- p9, l26: what exactly is the polynomial that you use. Include the co-efficients.

- p9, l28: delete 'Note that'

- p9, l31-32: include some statistics about the relative agreement before and after airmass correction.

- p10, l4: Please edit the start of the sentence here to make it objective.

- p10, l8: capitalize figure

- p10, l15: add a comma before which

- p10, l26: pointed out → showed

- p11, l1: I believe this campaign is now scheduled for 2017.

- p11, l4: Add 'The' before 'Aim'

- p11, l8: move also to before using

- p11, l13: delete the 2nd and 3rd instances of 'for'

- p11, l14: delete 'at'

- p16, l4-7: this reference is in a different format to the others

- Tables 1, 2: these could be consolidated into one table with a clear break at modification time

- Figure 4: maybe label the primary and secondary detectors in the Figure

- Figure 8: the numbers currently presented here give the impression that the ratios are different after the modification. I suggest including a measure of the uncertainty during each period, and an appropriate number of significant figures

- Figure 9, 10, 11: each of these figures would benefit from including a panel with residuals/differences.

- Figure 12: suggest changing the x-axis date to a more easily relatable format (YYYYMMDD or similar).

**3  References**

Butz, A., S. Guerlet, D. J. Jacob, D. Schepers, A. Galli, I. Aben, C. Frankenberg, J.-M. Hartmann, H. Tran, A. Kuze, G. Keppel-Aleks, G. C. Toon, D. Wunch, P. O. Wennberg, N. M. Deutscher, D. W. T. Griffith, R. Macatangay, J. Messerschmidt, J. Notholt, and T. Warneke (2011), Toward accurate CO2 and CH4 observations from GOSAT, Geophysical Research Letters, 38(14), 2-7, doi:10.1029/2011GL047888. Available from: http://www.agu.org/pubs/crossref/2011GL047888.shtml

Morino, I., O. Uchino, M. Inoue, Y. Yoshida, T. Yokota, P. O. Wennberg, G. C. Toon, D. Wunch, C. M. Roehl, J. Notholt, T. Warneke, J. Messerschmidt, D. W. T. Griffith, N. M. Deutscher, V. Sherlock, B. J. Connor, J. Robinson, R. Sussmann, and M. Rettinger (2011), Preliminary validation of column-averaged volume mixing ratios of carbon dioxide and methane retrieved from GOSAT short-wavelength infrared spectra, Atmospheric Measurement Techniques, 4(6), 1061-1076, doi:10.5194/amt-4-1061-2011. Available from: http://www.atmos-meas-tech.net/4/1061/2011/

Reuter, M., H. Bovensmann, M. Buchwitz, J. P. Burrows, B. J. Connor, N. M. Deutscher, D. W. T. Griffith, J. Heymann, G. Keppel-Aleks, J. Messerschmidt, J. Notholt, C. Petri, J. Robinson, O. Schneising, V. Sherlock, V. Velazco, T. Warneke, P. O. Wennberg, and D. Wunch (2011), Retrieval of atmospheric CO2 with enhanced accuracy and precision from SCIAMACHY: Validation with FTS measurements and comparison with model

results, Journal of Geophysical Research, 116(D4), 1-13, doi:10.1029/2010JD015047. Available from: http://www.agu.org/pubs/crossref/2010JD015047.shtml

Wunch, D., P. O. Wennberg, G. C. Toon, B. J. Connor, B. Fisher, G. B. Oster-man, C. Frankenberg, L. Mandrake, C. W. O'Dell, P. Ahonen, S. C. Biraud, R. Castano, N. Cressie, D. Crisp, N. M. Deutscher, A. Eldering, M. L. Fisher, D. W. T. Griffith, M. Gunson, P. Heikkinen, G. Keppel-Aleks, E. Kyrö, R. Lindenmaier, R. Macatangay, J. Mendonca, J. Messerschmidt, C. E. Miller, I. Morino, J. Notholt, F. A. Oyafuso, M. Rettinger, J. Robinson, C. M. Roehl, R. J. Salawitch, V. Sherlock, K. Strong, R. Suss-mann, T. Tanaka, D. R. Thompson, O. Uchino, T. Warneke, and S. C. Wofsy (2011), A method for evaluating bias in global measurements of CO2 total columns from space, Atmospheric Chemistry and Physics, 11(23), 12317-12337, doi:10.5194/acp-11-12317-2011. Available from: http://www.atmos-chem-phys.net/11/12317/2011/
* * *

---

## Referee Comment (RC2) · Anonymous Referee #2 · 4 Apr 2016

The paper describes the capacity of measuring XCO by adding a second detector channel to the existent portable FTIR spectrometer (EM27/SUN). Simultaneous monitoring of XCO with XCO2 and XCH4 is relevant for satellite validation and sources attribution.

Although the paper well describes the new instrumental set-up, results do not support the conclusions. The results need to be enriched by more quantitative and thorough comparisons between the prototype instruments and other reference sensors. Descriptions of comparisons should be strengthened: instead of comparing time series,

[Figure]

I would recommend using 1:1 (or scatter plots type) plots for more clarity and quantifications of the results. Section 5 should be improved and organized in sub-sections for more clarity.

I recommend publication of this paper in the AMT journal when comments are addressed.

- Why only using 6 days for the comparisons? Is it statistically adequate? Please comment.

- Use scatter plots (or other) for Fig 8, 9, 11, and 12, instead of tome series. Why the TCCON data are not used as a reference for such a validation. These data should be added to the comparison plots. Why not comparing XCO2 as well?

- Avoid the word "excellent agreement" in the actual comparisons.

- What is the precision of the prototype XCO, XCH4, and XCO2? How these precision compared to standard EM27/SUN?

- In figure 11, the agreements for last 3 days are different than the others days. Could you explain?

- Figures 5 and 6 can be combined together.

- Table 1 and 2 should be re-organized in one Table.

- Figure 7, could you add the interfering species in the window?

- Figure 9 shows the total column of CH4, not XCH4.

---

## Author Comment (AC1) · 28 Apr 2016

Dear referee#1: we have attached our responses and the revised manuscript (with and w/o tracking of changes) as supplements to this reply. Yours Sincerely, Frank Hase

Please also note the supplement to this comment:
http://www.atmos-meas-tech-discuss.net/amt-2015-403/amt-2015-403-AC1-supplement.zip